# Mental Health and Relations: Detection of Mental Health Disorders Related to Relationship Issues Through Reddit Posts

Leela Prasad Gorrepati
Camelot Integrated Solutions INC.
Virginia, United States
leelaprasad.gorrepati@gmail.com

Raj Sonani
Cornell University
New York, United States
sonaniraj@gmail.com

Venkatesh Velugubantla
Meridian Cooperative
Georgia, United States
venki.v@gmail.com

Ravi Teja Potla
Slalom
Texas, United States
raviteja.potla@gmail.com

MSVPJ Sathvik
AI Researcher
Hyderabad, India
msvpjsathvik@gmail.com

## Abstract

Relationship challenges, such as breakups and marital conflicts, often lead to mental health issues like depression and suicidal thoughts. To address this, we present a new binary classification dataset sourced from Reddit posts, specifically aimed at detecting mental health concerns triggered by relationship-related stress. The dataset categorizes text as either indicating mental health distress or not, offering valuable insights into the emotional effects of relationship problems. By applying advanced natural language processing techniques, this dataset supports early detection and intervention efforts. Furthermore, it can serve as a useful resource for government and non-government organizations (GOs and NGOs) to provide guidance and relationship counseling, offering timely support to individuals in need. The classifiers achieved an accuracy score of 78.43%.

## CCS Concepts

• **Computing methodologies** → **Natural language processing**.

## Keywords

Natural Language Processing, Social Media, AI for Social Good, Mental health, Relationship issues

**ACM Reference Format:**
Leela Prasad Gorrepati, Raj Sonani, Venkatesh Velugubantla, Ravi Teja Potla, and MSVPJ Sathvik. 2025. Mental Health and Relations: Detection of Mental Health Disorders Related to Relationship Issues Through Reddit Posts. In *Companion Proceedings of the ACM Web Conference 2025 (WWW Companion '25), April 28-May 2, 2025, Sydney, NSW, Australia.* ACM, New York, NY, USA, 5 pages. https://doi.org/10.1145/3701716.3717742

## 1 Introduction

Relationship issues, including breakups, marital conflicts, and other interpersonal problems, are common sources of emotional distress.

These issues can significantly impact mental health, often leading to conditions such as depression, anxiety, and suicidal ideation. The rising prevalence of mental health problems related to relationship stress underscores the urgent need for effective detection and intervention mechanisms.

Traditional methods of identifying mental health issues rely heavily on self-reporting and clinical evaluations, which can be time-consuming and inaccessible to many individuals. In recent years, the proliferation of social media and online forums has provided a rich source of data reflecting real-time emotional and psychological states. Platforms like Reddit, where users often share personal experiences and seek advice, offer valuable insights into the impact of relationship issues on mental health.

On Reddit, users frequently share relationship issues due to the platform's provision of anonymity, which ensures their identities remain concealed. This anonymity fosters a sense of security, allowing users to discuss intimate and personal matters without fear of being recognized or judged by acquaintances. Reddit's dedicated relationship subreddits, such as r/relationship_advice, offer supportive communities where individuals can receive diverse perspectives and immediate feedback. These communities are often moderated to maintain a non-judgmental atmosphere, encouraging open and honest communication. Additionally, the platform's vast user base provides empathy and advice from shared experiences, making it a valuable resource for those seeking understanding and solutions for their relationship challenges.

While Reddit offers valuable online support through its anonymous and diverse community, it is crucial to recognize that some individuals experiencing severe mental distress require in-person professional intervention. Online forums can provide immediate empathy and advice, but they lack the personalized, structured, and sustained support that trained mental health professionals can offer. In-person sessions allow therapists to observe non-verbal cues, build deeper therapeutic relationships, and create tailored treatment plans, which are essential for effectively addressing complex mental health issues. In addition, face-to-face therapy can provide a safe, confidential environment that encourages deeper emotional exploration and healing, ensuring that those in critical need receive the comprehensive care necessary for their well-being.

If NGOs and government organizations can access data from platforms like Reddit, they can use it to identify individuals in need and tailor their in-person sessions for more effective mental health

interventions. This data-driven approach can enhance the accuracy and relevance of their support, ultimately leading to better outcomes in curing and managing mental health issues. By leveraging online insights, these organizations can bridge the gap between digital support and essential face-to-face therapy. If there is an automatic algorithm which can find out who are in mentally ill it will be easier for the organisations for target helping.

The key contributions of our work is as follows:

(1) As of our knowledge we are the first to develop a dataset for mental health detection during breakups and relationship issues.
(2) The baseline classifiers have achieved an accuracy score of 78.43%.
(3) The dataset can be translated into different languages and used to build machine learning models in those respective languages as well.

## 2 Related Works

Several studies have explored the use of natural language processing (NLP) and machine learning (ML) techniques in detecting and analyzing mental health issues from social media data. Skaik and Inkpen [13] conducted a comprehensive review on using social media for mental health surveillance, emphasizing the potential of these platforms for early detection and intervention.

Gkotsis et al. Gkotsis et al. [5] examined the language patterns indicative of mental health problems in social media, highlighting the importance of computational linguistics in understanding psychological states. Le Le Glaz et al. [8] conducted a systematic review focusing on ML and NLP applications in mental health, outlining various approaches and their effectiveness.

Zhang et al. [17] provided a narrative review specifically on NLP applications for mental health detection, discussing advancements, challenges, and future directions in the field. Vajre et al. [16] introduced PsychBERT, a specialized language model tailored for analyzing mental health behaviors in social media contexts.

Inamdar et al. [6] explored ML-driven mental stress detection using NLP on Reddit posts, demonstrating the feasibility of automated monitoring and intervention. Bauer et al. [2] utilized large language models to understand suicidality and mental health disorders in social media discussions, contributing insights into linguistic markers and behavioral patterns.

Low et al. [10] conducted an observational study during COVID-19, revealing increased health anxiety and identifying vulnerable mental health support groups on Reddit through NLP analysis. Tadesse et al. [14] focused on the detection of depression-related posts in Reddit forums, showcasing the application of NLP for identifying mental health concerns in online communities.

Gaur et al. [4] contextualized classification of Reddit posts to DSM-5 criteria, proposing a framework for web-based interventions targeting mental health issues. These studies collectively illustrate the breadth of NLP and ML applications in understanding, detecting, and addressing mental health challenges using social media data.

Previous studies have concentrated on mental health in social media across different contexts and applications, yet they have overlooked the impact of relationships and breakups, which can profoundly affect mental health, potentially leading to suicidal ideation and depression.

## 3 Methodology

### 3.1 Data Collection and Annotation

We have collected the data from Reddit using selenium chrome webdriver. Selenium webdriver is an automatic web scraping tool. The selenium webdriver is programmed to automatically extract the text in the posts. The subreddit used for this are r/relationship_advice, r/relationships, r/BreakUps, r/RelationshipIndia, r/BreakUp, r/relationship_advicePH, r/breakupbuddy.

The annotation team consists of three interns who are proficient in English, having attended English-medium schools since high school and consistently achieving first-class marks in English. The team also includes two psychologists, one with around 6 years of experience and the other with over 4 years. The psychologists train the three interns in data annotation and review both their annotations and overall performance. They provide continuous feedback and guidance to help the interns improve their annotation skills.

In the annotation process, each data point is labeled by two different annotators. If both annotators assign the same label, that becomes the final label. If their labels differ, the data point is discussed with the psychologists, who then determine the final label. The annotators has to choose one of the three options mentally ill because of relationship issues(1), mentally ill because of not a relationship issue or not mentally ill(0) and the last one is they can opt psychologists help if they find it harder to find out the cause.

**What is mental illness or mental disturbance:** Mental illness or disturbance refers to conditions that affect a person's thinking, mood, or behavior in a way that disrupts their daily life. For instance, posts expressing suicidal thoughts, depression, or feelings of hopelessness are indicators of such disturbances and should be taken seriously as signs of underlying mental health issues.

**Guidelines:** Familiarize yourself with the data by thoroughly reviewing the dataset before beginning the annotation process to ensure consistent and accurate labeling. Regularly communicate with fellow annotators and psychologists, attending all training and review sessions to stay aligned with project standards. Maintain data confidentiality and follow security protocols. Use the provided Excel template, labeling each data point independently without prior discussion with other annotators. Psychologists will review conflicting labels and provide final decisions. Keep detailed notes on issues or uncertainties, and submit your annotated Excel sheet and notes at the end of each session.

We calculated inter-annotator agreement using Krippendorff's Alpha score metric [7]. For four annotators (a, b, c), the pairwise scores were computed as follows: $\alpha_{ab} = 0.793$, $\alpha_{ac} = 0.835$ and $\alpha_{bc} = 0.829$. The overall agreement score, $\alpha_t = 0.819$, was determined by averaging these pairwise scores.

### 3.2 Step-by-Step Guidelines for Annotators

*Familiarize Yourself with the Dataset.* Before starting the annotation, thoroughly review the dataset to understand the types of posts and context. This will help you maintain consistency and accuracy throughout the process.

**Table 1: Overview of the dataset**

| Input | Output |
|---|---|
| fuck...i recently went through a breakup with my partner of three years, been the most devastating experience of my life. We were so deeply connected, and I thought we had a future together. am struggling to get through each day, replaying all the moments we shared. The pain is so intense that I've lost interest in everything I once loved, and I've even started having suicidal thoughts. | 1 |
| Allow yourself to feel the full spectrum of emotions that come with a breakup—sadness, anger, confusion. Embracing these feelings with acceptance can lead to emotional healing and eventual closure. | 0 |
| My partner left me a month ago, and I've been spiraling ever since. We were together for so long, and now I feel like a part of me is missing. The depression has gotten so bad that I've started thinking about ending it all. I feel like I'm trapped in this never-ending cycle of pain and despair, and I'm scared of where my thoughts are taking me. If anyone has been through something similar, how did you cope? I really need some support right now. | 1 |
| Relationships are often perceived as mere vessels for lust, devoid of any meaningful benefits or genuine help. They can lead to heartbreak, disappointment, and wasted time. Instead of investing in relationships, focus on personal growth and individual pursuits that truly matter. Self-reliance and independence offer more satisfaction and stability than relying on fleeting romantic entanglements. Prioritize building a strong sense of self-worth and purpose rather than seeking validation or fulfillment through relationships that may not endure. Remember, true happiness comes from within and from nurturing meaningful connections beyond romantic pursuits. | 0 |

**Table 2: Data Statistics. 1 represents mentally disturbed due to relationship, 0 represents mentally disturbed due to other reasons or no mental illness**

| Metric | 0 | 1 | Overall |
|---|---|---|---|
| Data Points | 7859 | 7748 | 15607 |
| Number of Words | 80213 | 79045 | 159258 |
| Word density | 10.21 | 10.20 | 10.20 |

**Table 3: Test results: Detection of mental health. ft(Finetuning), fs(Few Shot) and z(Zero Shot)**

| Model | p | r | A |
|---|---|---|---|
| BERT(ft) | 68.52 | 69.39 | 68.82 |
| DistilBERTa(ft) | 67.39 | 66.03 | 67.51 |
| RoBERTa(ft) | 62.48 | 61.25 | 63.70 |
| LLAMA 3(z) | 51.42 | 52.73 | 51.86 |
| LLAMA 3(fs) | 58.49 | 57.63 | 57.66 |
| LLAMA 3(ft) | 71.36 | 68.83 | 70.52 |
| Gemini(z) | 50.29 | 52.30 | 51.30 |
| Gemini(fs) | 56.46 | 58.21 | 57.39 |
| GPT-4o(z) | 63.71 | 64.33 | 63.84 |
| GPT-4o(fs) | 68.30 | 67.94 | 68.27 |
| GPT-4o(ft) | 77.69 | 79.42 | 78.43 |

*Independent Labeling.*

- Label each post independently. Do not discuss your annotations with other annotators during the initial labeling phase.
- Use the provided Excel template for this process.

*Handling Disagreements.*

- Each post is labeled by two different annotators. If both annotators assign the same label, that becomes the final label.
- In cases of disagreement, the psychologists will review the post and make the final decision after discussion with the annotators if necessary.

*Confidentiality.* Ensure all data is handled confidentially and follow the prescribed security protocols.

*Attend Training and Feedback Sessions.*

- Regularly attend training and review sessions with the psychologists to ensure your understanding of mental health issues remains aligned with the project requirements.
- Continuous improvement is expected based on the feedback provided.

*Documentation.*

- Keep detailed notes on any uncertain or difficult cases. These will be discussed during the psychologist review sessions.
- At the end of each session, submit your annotated Excel sheet along with any notes taken.

*3.2.1 Statistics.* Table 2 provides dataset statistics related to mental illness, categorized by whether the illness is due to relationship issues (represented by 1) or other reasons/no mental illness (represented by 0). There are a total of 15,607 data points, with 7,748 falling into the relationship-related mental illness category and 7,859 in the other category. The total number of words across the dataset is 159,258, with 79,045 words associated with the relationship category and 80,213 words in the other category. The word density, calculated as the number of words divided by the number of data points, is very similar across both categories at around 10.20 words per data point overall. Table 1 presents the overview of the dataset.

## 3.3 Baselines

The following models are used for baseline implementations: BERT [3], RoBERTa [9], DistilBERT [12], Gemini [15], GPT-4o [11], and LLAMA 3 [1]. The GPT-4o model is accessed via the OpenAI GPT-4o API through the OpenAI Playground, while the Gemini model utilizes the Google Generative API from Google Cloud. The remaining models are implemented using the Hugging Face library.

For fine-tuning the baselines, the learning rate was set between 0.00002 and 0.00005, depending on the specific model. We utilized a batch size of 16 and a maximum sequence length of 128 tokens for all models. The AdamW optimizer was employed, with a weight decay rate of 0.01 to reduce overfitting, and the learning rate was adjusted using a linear scheduler with 10% warm-up steps of the total training iterations. Fine-tuning was carried out for three epochs, but early stopping was applied if no improvement in validation loss was observed for two consecutive epochs. Additionally, gradient clipping was set to 1.0 to prevent gradient explosion during training, and dropout was applied with a probability of 0.1 on fully connected layers to further mitigate overfitting. Mixed precision training was enabled to optimize memory usage and reduce training time. To evaluate the model's performance, precision, recall, F1-score, and accuracy metrics were recorded after each epoch.

Three techniques are employed: zero-shot (z), few-shot (fs), and fine-tuning (ft). Zero-shot learning uses no examples, few-shot learning uses a small number of examples, and fine-tuning involves training on the entire training dataset. The dataset is divided into 80% for training and 20% for testing. To detect mental states, we utilized precision, recall, and accuracy as our evaluation metrics.

## 4 Experimental Results and Discussion

The results show that GPT-4o achieved the highest scores across all three metrics, with 63.71 for ft(Finetuning), 64.33 for fs(Few Shot), and 63.84 for z(Zero Shot). GPT-4(z) closely followed with scores of 63.39, 62.30, and 79.42 respectively. These high scores indicate the strong performance of the GPT-4 models in detecting Breakup Blues. BERT and DistilBERT also performed reasonably well, with BERT scoring 68.52, 69.39, and 68.82, slightly higher than DistilBERT()'s scores of 67.39, 66.03, and 67.51 for the three metrics. These results suggest that both models have good capabilities for this task, with BERT having a slight edge.

Gemini had the lowest scores among the models listed, with 50.29, 50.24, and 58.21 for the three metrics. These results demonstrate the superior performance of GPT-4 and the strong capabilities of BERT and DistilBERT for the task of detecting Breakup Blues. The LLAMA models and Gemini exhibited relatively lower performance, but their scores varied across the different metrics.

The practical applications of the classifiers are as follows:

**Targeted Intervention by Mental Health Professionals:** Focus resources on individuals identified by classifiers to provide specialized assistance for breakup-related mental health challenges.

**Educational and Awareness Campaigns:** Develop campaigns informed by classifier data to educate about coping strategies and reduce stigma around breakup-related mental health challenges.

**Social Media Monitoring and Support:** Automatically identify and reach out to users displaying mental distress post-breakup for timely intervention.

**Tele Support:** Psychology students can use the classifiers to detect mentally ill users and provide remote counseling.

## 5 Conclusion

In this study, we developed classifiers trained on the proposed dataset to detect mental health issues related to relationship issues. The classifiers achieved 78.43% accuracy in identifying posts indicating mental distress following relationship problems. This highlights the potential of natural language processing and machine learning in recognizing emotional states on social media.

## Limitations

The study is constrained by its focus on English-language content, which may restrict the generalizability of findings to non-English-speaking populations and cultures where social media use and expression patterns differ significantly. While the dataset predominantly covers mental health issues related to romantic and marital relationships, it may underrepresent or omit other significant relationship dynamics such as parent-child relationships (e.g., father-son, mother-son). This limitation may skew the findings towards specific relationship contexts, potentially overlooking important insights and challenges faced in familial relationships.

## Ethics Statement

We prioritize safeguarding user privacy by refraining from making user IDs public. This precaution is essential because publicly releasing such information can raise significant privacy concerns and increase the risk of misuse that could potentially lead to social harm. We are firmly committed to preventing such misuse and do not condone any actions that could compromise user privacy or contribute to societal harm. Our primary objective is to advance AI technology for the betterment of society. By maintaining strict privacy standards and advocating responsible data handling practices, we aim to ensure that our contributions to AI innovation are aligned with promoting social good.

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
