# OpenReview forum: "Mental Health and Relations: Detection of Mental Health Disorders Related to Relationship Issues Through Reddit Posts"
_ACM.org/TheWebConf/2025/Workshop/TIME — TIME 2025 Poster_

### Official Review · Reviewer_eYJg · 2025-01-10
**The work presents a novel dataset for detecting mental health issues related to relationship stress using Reddit posts, employing NLP and ML techniques. It is well-structured, with a clear annotation process involving psychologists, ensuring reliability. The study is original, addressing a significant gap by focusing on relationship-related mental health issues. It has practical significance for real-world applications in mental health interventions. However, the dataset may not fully represent all relationship dynamics, and the classifier's accuracy of 78.43% indicates room for improvement. Overall, it is a valuable contribution to mental health research.**

**Rating:** 7
**Confidence:** 3

**Review:**

Quality:

The work appears to be well-structured, focusing on the development of a dataset for detecting mental health issues related to relationship stress using Reddit posts. The methodology includes the use of natural language processing (NLP) and machine learning (ML) techniques, which are appropriate for the task.
The study includes a clear process for annotating data, involving independent labeling and resolution of disagreements by psychologists, which adds to the reliability of the dataset.

Clarity:

The document clearly outlines the motivation behind the study, emphasizing the impact of relationship issues on mental health and the potential of social media data for early detection.
The guidelines for annotators are detailed, ensuring that the process is transparent and can be replicated or reviewed by others.

Originality:

The work claims to be the first to develop a dataset specifically for mental health detection during breakups and relationship issues, which suggests a novel contribution to the field.
It leverages social media data, which is a relatively new and innovative approach in mental health research.

Significance:

The study addresses a significant gap in mental health research by focusing on relationship-related stress, which is a common but often overlooked area.
The potential application of the dataset for government and non-government organizations to provide timely mental health support highlights its practical significance.

Overall Pros:

Novel dataset focused on a specific and impactful area of mental health.
Use of advanced NLP and ML techniques for data analysis.
Clear and structured annotation process with expert involvement.
Potential for real-world application in mental health interventions.

Overall Cons:

The dataset may underrepresent other significant relationship dynamics, such as familial relationships, which could limit the generalizability of the findings.
The accuracy score of 78.43% for the classifiers, while decent, suggests there is room for improvement in the model's performance.

Overall, the work is a valuable contribution to the field of mental health research, offering new insights and practical applications, though it could benefit from addressing its limitations in future studies.

---

### Official Review · Reviewer_PTjL · 2025-01-13

**Rating:** 5
**Confidence:** 3

**Review:**

The author developed classifiers trained on the proposed dataset to detect mental health issues related to relationship issues. I think this paper is well-structured and easy-to-follow.

Strengths \
For me, I think this problem is interesting. Mental health is an important issue, research this via reddit. I think it's interesting. \
The process of data collection is clear.

Weakness \
Wouldn't a direct LLM classification have done the job? And the labeling with LLM is not always correct. It's a little strange.

---

### Official Review · Reviewer_iuML · 2025-01-13
**Reviews of Mental Health and Relations: Detection of Mental Health Disorders Related to Relationship Isuues Through Reddit Posts**

**Rating:** 6
**Confidence:** 4

**Review:**

The paper introduces a dataset that captures content related to mental health concerns from Reddit posts. The dataset is annotated by human labelers, with labels being 0/1, where 1 represents being mentally disturbed due to relationship issues, and 0 represents being mentally disturbed due to other reasons or no mental illness. The classification achieved an accuracy of 78.43% using GPT-4o.

Advantages:

- The research topic holds positive significance for addressing mental health issues caused by relationship problems.
- The dataset proposed in the paper is valuable.
- The paper evaluates the prediction results of different models.

Disadvantages:

- Did the authors attempt to fine-tune GPT-4o to achieve better predictive performance?

---

### Official Review · Reviewer_EckR · 2025-01-20
**Nice idea but need address critical concern about Data Collection**

**Rating:** 5
**Confidence:** 5

**Review:**

This paper presents an innovative approach to detecting mental health issues related to relationship problems through the analysis of Reddit posts. The study's focus on developing a dataset and classifiers for this specific purpose is commendable and addresses an important gap in mental health research. However, there are several areas where the paper could be strengthened to enhance its impact and address potential concerns.

The methodology section provides a detailed description of the data collection and annotation process, which is a strength of the paper. The use of multiple annotators, including psychologists, adds credibility to the dataset. However, the inter-annotator agreement scores, while good, could be improved. Future work might consider refining the annotation guidelines or providing more extensive training to annotators to increase agreement.

***A critical concern that must be addressed is the ethical and legal implications of the data collection method. Reddit's User Agreement explicitly prohibits unauthorized scraping or data collection, including "using any robot, spider, crawler, scraper, or other automated means or interface to access our Services or extract other users' information." The use of Selenium for web scraping, as described in the paper, likely does not comply with Reddit's terms of service. This raises significant ethical and potentially legal issues that could undermine the validity and usability of the research.***

```
To address this, the authors should:
1. Obtain explicit authorization from Reddit for their data collection methods.
2. Consider using official API channels provided by Reddit for data collection.
3. Thoroughly review and comply with Reddit's policies and terms of service.
4. Clearly document in the paper the steps taken to ensure compliance with Reddit's policies.

Helpful Links for reference:
- https://www.reddit.com/r/reddit/comments/1co0xnu/sharing_our_public_content_policy_and_a_new/
- https://redditinc.com/policies/user-agreement
```

The experimental results demonstrate promising performance of the classifiers, particularly the GPT-4o model, which achieved 78.43% accuracy. This is a solid foundation for future research in this area. However, the paper would benefit from a more in-depth discussion of why certain models performed better than others and what implications this has for future model development in mental health detection.

The conclusion section, while highlighting the main achievement of the study, lacks a comprehensive discussion of the broader implications of the findings. The authors should expand on how these classifiers could be applied in real-world mental health interventions and what challenges might arise in their implementation. Additionally, specific suggestions for future research directions, such as improving classifier accuracy or expanding to other types of relationships, would strengthen the paper's contribution to the field.

The limitations section acknowledges some important constraints of the study, such as its focus on English-language content and romantic relationships. However, it does not adequately address potential biases in the data collection and analysis process. The authors should discuss how the demographics of Reddit users might skew the results and consider the impact of cultural differences on expressing mental health issues online. Additionally, the potential biases introduced by the annotation process and how these were mitigated should be explored.

In conclusion, while the paper makes a valuable contribution to the field of mental health research using natural language processing, significant revisions are necessary. The authors must address the critical issue of data collection compliance with Reddit's policies, expand on the limitations and ethical considerations, and provide a more comprehensive discussion of the implications and future directions of this research. With these revisions, this paper has the potential to make a significant impact in the field of mental health detection using social media data, providing a solid foundation for future studies aimed at leveraging AI technologies for early detection and intervention in mental health issues related to relationship problems.

---

### Meta-Review · Area_Chair_8iai · 2025-01-25

**Recommendation:** Accept (Poster)
**Confidence:** 4

**Metareview:**

The paper takes on a really important topic, using AI to detect mental health issues caused by relationship stress, and brings something unique to the table with its new dataset. The annotations done by psychologists add a lot of credibility. The models hit a solid accuracy of 78.43%, but it would’ve been great to see more diverse data or extra features beyond text classification to back up the claims.

Clarity: The writing is clear and easy to follow, and the methodology is broken down step by step, which makes it approachable. The tables showing dataset stats and model performance are really helpful too. That said, the section on model configurations gets a bit technical and could use some simplifying for readers less familiar with the details.
Originality: The focus on relationship-related mental health issues is a fresh take, and the dataset built around this specific theme is a valuable contribution. Using Reddit as the data source makes sense and feels relevant. However, while the dataset is new, the models and techniques like GPT-4 and BERT are pretty standard and don’t bring anything particularly groundbreaking.

Pros: The dataset is a solid addition to this area of research, and it’s great that trained psychologists were involved in the annotation process. The performance of the models, especially GPT-4, is impressive for this type of task. The paper also does a good job of pointing out real-world uses like targeting mental health interventions.
Cons: The dataset is limited to English, so it might not work as well in other languages or cultural settings. Also, while the paper talks about practical applications, there aren’t any concrete examples or real-world case studies to show how this could work outside a research setting.

Overall: The paper addresses a meaningful problem and introduces a valuable dataset, but it feels more like a starting point than a finished solution. It’s a great fit for a poster presentation where it can generate discussions and ideas for future work.

---

### Decision · Program_Chairs · 2025-01-26

**Decision:**

Accept (Poster)

**Comment:**

The program chair concurs with the area chair's decision.

For the camera-ready version, please revise your paper according to the feedback provided by the reviewers.

Workshop papers must be written in English, follow a double-column format, and comply with the [ACM template](https://www2025.thewebconf.org/short-papers) and formatting guidelines. The template is also available in [Overleaf](https://www.overleaf.com/latex/templates/association-for-computing-machinery-acm-sig-proceedings-template/bmvfhcdnxfty). For authors using Microsoft Word, the Word Interim Template is recommended.

Camera-ready versions of accepted papers can and should include all information to identify authors, and should acknowledge any funding received that directly supported the presented research.

In addition, ensure that the DOI (to be provided by the PCs at a later stage) is included, and cite the workshop (to appear) using the following reference:

```
@inproceedings{time2025,
  title={TIME 2025: 1st International Workshop on Transformative Insights in Multi-faceted Evaluation},
  author={Lei Wang and Md Zakir Hossain and Syed Islam and Tom Gedeon and Sharifa Alghowinem and Isabella Yu and Serena Bono and Xuanying Zhu and Gennie Nguyen and Nur Haldar and Seyed Jalali and Abdur Razzaque and Imran Razzak and Rafiqul Islam and Shahadat Uddin and Naeem Janjua and Aneesh Krishna and Manzur Ashraf},
  booktitle={ACM Web Conference Workshop},
  year={2025}
}
```

Please note that at least one in-person registration is required for each accepted workshop paper to be included in the Companion Proceedings of WWW 2025. All accepted papers must be presented at the conference. Papers not presented (no-shows) may be withdrawn from the companion proceedings. Presentations will be conducted in two formats: oral and poster.

The camera-ready deadline for workshop papers is 7 February 2025 (AoE).